# Bridging the Gap in Cancer-Related Stroke Management: Update on Therapeutic and Preventive Approaches

**DOI:** 10.3390/ijms24097981

**Published:** 2023-04-28

**Authors:** Ming-Yee Sun, Sonu M. M. Bhaskar

**Affiliations:** 1Global Health Neurology Lab, Sydney, NSW 2000, Australia; 2Neurovascular Imaging Laboratory, Clinical Sciences Stream, Ingham Institute for Applied Medical Research, Liverpool, NSW 2170, Australia; 3UNSW Medicine and Health, University of New South Wales (UNSW), South Western Sydney Clinical Campuses, Sydney, NSW 2170, Australia; 4Department of Neurology and Neurophysiology, Liverpool Hospital and South West Sydney Local Health District (SWSLHD), Liverpool, NSW 2170, Australia; 5NSW Brain Clot Bank, NSW Health Pathology, Sydney, NSW 2170, Australia; 6Stroke and Neurology Research Group, Ingham Institute for Applied Medical Research, Liverpool, NSW 2170, Australia

**Keywords:** cancer, stroke, thromboembolism, anticoagulation, prevention, reperfusion, cerebrovascular disorders

## Abstract

The underlying aetiopathophysiology of cancer-related strokes and thromboembolisms differs from that of noncancer-related strokes, which makes treating cancer-related strokes and thromboembolisms a distinct clinical challenge. This necessitates the development of novel, individualised diagnostic and treatment strategies. However, limited guidelines are available for the management of cancer-related strokes and the prevention of acute strokes or other thromboembolic events in this patient population. In this article, we present an updated overview of the therapeutic and preventive strategies for strokes in cancer settings. These strategies include acute reperfusion therapy, anticoagulant therapy, antiplatelet therapy, and lifestyle management options. We also outline comprehensive pathways and highlight gaps in the evidence-based clinical management of cancer-related strokes or thromboembolisms. Additionally, future recommendations for the management of strokes in cancer patients are provided.

## 1. Introduction

Cancer-related strokes have generated keen scientific and clinical interest [1,2]. They have a unique aetiopathophysiology distinct from noncancer-related strokes, warranting tailored/distinct treatment considerations [3]. Cerebrovascular disease (CVD) is frequently present in cancer patients, and there is a demonstrated link between cancer and stroke. Approximately, one in seven to eight ischaemic stroke patients has either occult or recognised cancer. Within this subgroup, cancer-related coagulopathy has been implicated as the underlying mechanism in 40% of ischaemic stroke patients with cancer [4,5,6]. In 1823, Jean-Baptiste Bouillaud [7], a French physician, published the first report on the association between cancer and stroke [8]. The landmark 1985 autopsy study by Graus et al. reported that the most common central nervous system complication in cancer patients, following metastasis, was cerebral infarction and haemorrhage, with pathologic evidence of CVD in 14.6% of autopsied cancer patients, out of which one in two cases of cancer with CVD were symptomatic [9]. Whilst the link between ischaemic stroke and cancer is well established [10], whether cancer patients are more likely to experience spontaneous intracerebral haemorrhage (ICH) is a subject of ongoing controversy [11].

Though there are a variety of prophylactic treatments and guidelines available for the management of stroke [12], research specific to the high-risk phenotype of cancer patients is limited, and therapies are only utilised and implemented to a limited degree due to a lack of awareness amongst clinicians regarding the risk of stroke in cancer [13]. This leads to unwarranted variations in clinical care in the management of the risk of thrombosis in cancer patients. Precision medicine-based approaches towards optimal primary and secondary prevention interventions may mitigate stroke risk and hence improve outcomes in cancer patients. Whilst guidelines exist for the secondary prevention of strokes and transient ischaemic attacks (TIA) [12], the guidelines specific to the management and preventive screening of stroke in cancer patients are limited [13]. Therefore, targeted strategies to identify subgroups of patients most at risk of a stroke are important for stroke prevention in cancer patients. This article presents a comprehensive review and updates on appropriate therapeutic management workflows for stroke in cancer as well as recommendations for future research and clinical practice improvement.

## 2. Therapeutic or Preventive Management of Stroke in Cancer Patients

Overall, amongst both cancer and noncancer patients, the incidence of ischaemic and nonischaemic strokes remain consistent at 85% and 15%, respectively [14]. The highest risk of an ischaemic stroke after cancer is within the first month of cancer diagnosis, with stage 4 cancer patients facing a 10-fold increased risk in this period compared to the normal population [15]. For haemorrhagic stroke, the risk was greatest in the first 6 months after diagnosis, at almost 2 times the normal population [16]. Another important clinical consideration is atrial fibrillation (AF) in cancer patients who develop stroke [10]. About 2–5% of cancer patients have AF at the time of diagnosis, making it yet another common condition in this population [17,18,19]. A case-controlled study of cancer patients who experience stroke/transient ischaemic attack (TIA) with matched cancer patients without stroke/TIA found that AF, prior ischaemic stroke, ongoing cancer therapy, dyslipidaemia, and renal disease are independent risk factors for stroke/TIA [19]. The study also indicated that a higher CHA_2_DS_2_-VASc score significantly potentiates the risk in people with active cancer independent of AF [19]. This study had several limitations including a retrospective design and a small sample size. Further studies on the interrelationship among AF, stroke, and cancer may provide insights into the management of this high-risk subgroup of patients. The assessment of the risk factors identified in this study may guide preventive strategies in cancer patients who develop stroke/TIA. Timely preventive and therapeutic management of stroke in cancer patients is essential in both improving their prognosis and in reducing the burden that these diseases individually and combined place on the health system. Cancer-specific stroke risk prediction scores/tools need to be developed to aid the prevention of stroke/TIA [19]. Strategies for the prophylactic and emergent management of stroke in cancer patients are discussed below.

## 3. Reperfusion Therapy for Stroke in Cancer Patients

Reperfusion therapy, intravenous thrombolysis (IVT), and endovascular thrombectomy (EVT) are currently the mainstay of acute management of ischaemic stroke patients [20]. Reperfusion therapy for stroke involves the administration of drugs to dissolve blood clots. The tissue plasminogen activator (tPA) is the most commonly used thrombolytic drug. The treatment with tPA works by dissolving the clot and restoring blood flow to the affected area of the brain. However, the use of tPA in cancer patients can be complicated by several factors including thrombocytopenia, impaired coagulation, and the need for additional monitoring. However, limited studies have been conducted on the effectiveness of IVT or EVT in stroke patients with active malignancy, and those that have been conducted have used small sample sizes [21,22,23,24,25]. Hence, treatment guidelines for stroke in cancer patients are not well established. The 2019 guidelines from the American Heart Association and American Stroke Association suggest that acute ischaemic stroke patients with systemic cancer and reasonable (>6 months) life expectancy may benefit from IVT [26]. No such recommendations are available for EVT in patients with acute ischaemic stroke and active cancer [27]. A substudy of the Multicenter Randomized Clinical Trial of Endovascular Treatment for Acute Ischaemic Stroke in the Netherlands (MR CLEAN) Registry provided Class I evidence that patients with active cancer undergoing EVT for acute ischaemic stroke (AIS) had significantly worse functional outcomes at 90 days relative to those without active cancer. Authors also reported an increased risk of recurrent stroke in cancer patients [23]. A study by Joshi et al. found that the functional outcomes and mortality at 90 days of patients following EVT for AIS with and without cancer were similar in a propensity-matched analysis [22]. Cancer patients, on the other hand, had a much greater risk of haemorrhagic transformation (HT) [22]. An analysis of the SECRET (Selection Criteria in Endovascular Thrombectomy and Thrombolytic Therapy) registry on 1338 patients who underwent reperfusion therapy, comprising 62 patients (4.6%) with active cancer, 78 patients (5.8%) with nonactive cancer, and 1198 patients (89.5%) with no history of cancer revealed similar adverse events and 24 h neurological improvement in patients with active cancer relative to other groups [28]. However, patients with active cancer were linked to poorer long-term functional outcomes, vis-à-vis functional independence at 3 months and mortality at 6 months. A systematic review comprising 18 retrospective studies on the safety and efficacy of MT in cancer patients revealed EVT is safe in cancer patients with acute ischaemic stroke, however, with higher mortality at 90 days and lower 90-day functional independence [29]. Another meta-analysis, aided by machine learning, on the safety and outcomes of reperfusion therapy, IVT or EVT in AIS patients with or without active cancer reported comparable clinical outcomes of IVT and procedural outcomes of EVT, indicating reperfusion therapy may benefit a select group of AIS patients with active cancer [30]. However, IVT was associated with an increased risk of symptomatic intracerebral haemorrhage (sICH) in AIS patients with active cancer. Furthermore, in active cancer patients treated with EVT for AIS, higher mortality and fewer odds of favourable outcomes were observed [30]. Interestingly, a recent study leveraging the National Inpatient Sample (NIS), a large inpatient database from the United States, showed that in contemporary medical practice settings, acute ischaemic stroke patients with comorbid cancer or metastatic cancer treated with EVT had similar rates of intracranial haemorrhage and favourable/routine discharges as patients without cancer [31]. However, patients with metastatic cancer were associated with significantly higher rates of in-hospital mortality compared to patients without cancer [31]. Indeed, the efficacy of EVT in cancer patients presenting with stroke merits further investigation. Furthermore, efforts to reduce treatment delays in providing time-critical reperfusion therapies to acute stroke patients, including those with pre-existing cancer, are also needed [32,33].

## 4. Anticoagulation

The preventive and therapeutic management of stroke in cancer patients is essential in improving prognosis and in reducing the burden that these diseases individually and combined place on the health system.

Cancer patients undergoing chemotherapy may be at increased risk of venous thromboembolism (VTE) [34,35]. Identifying patients at high risk of VTE is crucial for prophylactic treatment strategies, such as anticoagulation. In comparison to matched controls, patients with cancer-related stroke exhibit greater levels of coagulation, platelet, and endothelial dysfunction as well as more circulating microemboli [36]. A multicentre prospective cohort study of adult patients (N = 50) with AIS and cancer found that the primary outcome of a composite of recurrent arterial/VTE or death occurred in 43 (86%) of the individuals [37]. The composite of recurrent arterial/VTE or mortality was associated with the levels of D-dimer (Hazards Ratio (HR), 1.6), P-selectin (HR, 1.9), sICAM-1 (HR, 2.2), sVCAM-1 (HR, 1.6), and microemboli (HR, 2.2). Recurring AIS was linked to the D-dimer level (HR, 1.2). For the stroke-only group, only one biomarker (P-selectin) and for the cancer-only group, only thrombin–antithrombin, were shown to be uniquely linked. These findings imply that indicators of embolism and hypercoagulability may be linked to poor clinical outcomes in people with cancer-related AIS [37]. Further research on haematologic and embolic prognostic biomarkers is warranted. In clinical practice, a biomarker signature may need to take into account factors including unambiguous interpretability, testing cost, and convenience in addition to predictive accuracy [38]. Concentrating on a subset of well-characterised biomarkers may aid in minimising costs.

Two landmark randomised clinical trials (RCTs) indicated that VTE prophylaxis with direct oral anticoagulants (DOACs), following a risk assessment, significantly reduced the rate of VTE during chemotherapy [39,40]. The 2021 guidelines from the American Society of Hematology recommend stratification of cancer patients according to their VTE risk prior to chemotherapy, as well as considering patient-specific factors, using the Khorana risk score, which takes into account the cancer phenotype [41]. More recently, the 2022 International Clinical Practice Guidelines published by the International Initiative on Thrombosis and Cancer (ITAC) also provided indications for treatment and prophylaxis of cancer-associated thrombosis [42]. The following are key recommendations (grade 1A or 1B): (1) low-molecular-weight heparins (LMWHs) for initial (first 10 days) and maintenance therapy of cancer-associated thrombosis; (2) in the absence of significant drug–drug interactions or gastrointestinal absorption impairment, DOACs are indicated for the initial treatment and maintenance of cancer-associated thrombosis in patients who are not at elevated risk of gastrointestinal or genitourinary bleeding; (3) use of LMWHs or DOACs to treat cancer-related thrombosis for a minimum of 6 months; (4) prophylactic treatment with LMWHs for an extended period of time (4 weeks) to prevent postoperative VTE after major abdominopelvic surgery, if there is no significant bleeding risk; and (5) use of LMWHs or direct oral anticoagulants (apixaban or rivaroxaban) is indicated for primary prevention of VTE in ambulatory patients with locally advanced or metastatic pancreatic cancer undergoing anticancer treatment and have a low likelihood of bleeding [42].

The American Society of Oncology (ASCO) Clinical Practice guidelines also provide indications for prophylaxis and treatment of VTE in patients with cancer [43]. They indicate pharmacologic thromboprophylaxis to hospitalised patients with active malignancy in the absence of bleeding or other contraindications. Whilst routine pharmacologic thromboprophylaxis was not indicated to all outpatients with cancer, they recommended thromboprophylaxis with apixaban, rivaroxaban, or LMWH in high-risk outpatients with cancer (Khorana score of 2 or higher before starting a new systemic chemotherapy regimen). Furthermore, they also indicated pharmacologic thromboprophylaxis with either unfractionated heparin (UFH) or LMWH to all patients with malignant disease undergoing surgical intervention.

Patients with cancer and VTE should receive anticoagulation for a minimum of 6 months [44]. However, the National Comprehensive Cancer Network (NCCN) guidelines recommend a minimum duration of 3 months. In patients with cancer-associated VTE, a meta-analysis of randomised trials indicates a decreased risk of recurrent VTE albeit with a higher risk of bleeding compared to LMWH [45]. The 2021 Guidelines for the Secondary Prevention of Ischaemic Stroke released by the American Heart Association/American Stroke Association incorporates a section on malignancy, with recommendations being for patients with ischaemic stroke or TIA in the setting of AF and cancer, and DOACs are preferred over warfarin for stroke prevention [12]. They also called for action and highlighted the need for more specific guidelines on the type of anticoagulation across specific cancers and for how to treat patients who are on anticoagulation but still experience a stroke and the effect of LMWH. 

## 5. Antiplatelet Therapy

In the absence of an embolic source, antiplatelet therapy is indicated in patients with malignancy and ischaemic stroke [46]. Antiplatelets should also be considered as a preventive therapy, as they can decrease platelet aggregation and thus impede potential thrombus formation [15]. As platelets form the basis of the haemostatic plug, which adheres to vascular lesions, targeting these main effector cells of coagulation and thrombosis could be paramount [47]. Damaged endothelium secretes von Willebrand factor (vWF) and other thrombotic mediators, which activate platelet adhesion and aggregation. Furthermore, increased platelet counts are a risk factor for VTE in cancer patients, particularly those undergoing concurrent chemotherapy [48]. Agents, such as acetylsalicylic acid, clopidogrel, phosphodiesterase inhibitors, and glycoprotein IIb/IIIa inhibitors, are conventional therapies in the treatment of arterial thrombosis and may apply to the scenario of stroke in cancer patients, warranting further research [48].

For patients without a need for anticoagulation, antiplatelet therapy may be initiated. Anticoagulation is more appropriate for patients with a Factor V Leiden disorder, with a large-vessel dissection or AF (cardiogenic embolism/thrombophilic conditions). Antiplatelet therapy is preferred also in lesions that are characterised by atherosclerosis and endothelial injury [49]. Dual antiplatelet therapy (DAPT) may need to be discontinued in cancer patients who have received percutaneous coronary intervention (PCI) or experienced an acute coronary syndrome (ACS) to restart anticancer treatment, perform surgery, or even have biopsies [50,51]. Furthermore, the optimal duration of DAPT in cancer patients remains a challenge owing to the increased risks of thrombosis and haemorrhage associated with cancer [52]. By using optical coherence tomography (OCT) to assess the stent strut coverage, the decision around early DAPT discontinuation can be considered in high-risk patients [53]. 

Furthermore, histological studies into the molecular features of thrombi in cancer patients who experienced stroke reveal that they tend to be higher in platelet fraction and lower in erythrocyte in comparison to inactive cancer or control groups [54,55]. Patients with vegetation also showed high platelet and low erythrocyte fractions. This points to the importance and role of antiplatelet therapy in the context of cancer patients at a higher risk of stroke. Though there is still limited literature available in this area [56,57], more studies that focus on analysing thrombus and clot morphology in patients with stroke and cancer will inform the selection of the most efficacious treatment option.

## 6. Lifestyle Management

The identification of independent stroke risk factors in cancer patients is essential, given that the majority of patients in the studies observed above undergoing radiotherapy and chemotherapy experienced multiple confounding factors of hypertension, hyperlipidaemia, diabetes mellitus, and tobacco use [4]. Managing modifiable factors, such as hypertension, hyperlipidaemia, and diabetes, should be concurrent with any sort of pharmacological treatment, as well as lifestyle modifications, including regular physical activity, healthy diet, hydration, weight management, compression stockings, and smoking cessation counselling [58,59].

## 7. Discussion

Cancer-related stroke or thromboembolism is a significant and potentially life-threatening complication that can affect individuals with cancer [1,2]. When compared to the general population, people with cancer have a significantly higher risk of stroke or thromboembolism [15,44]. Certain phenotypes of cancer, such as lung, pancreatic, and brain cancer, carry an elevated risk of cancer-related stroke or thromboembolism [3]. Furthermore, cancer therapies, such as chemotherapy, radiation therapy, and surgery, might increase the risk of blood clots and strokes [1,35]. For instance, chemotherapy medications might cause damage to the lining of blood vessels, which can result in inflammation and clotting [60]. Radiation therapy has also been linked to an increased risk of blood clots due to its potential to damage blood vessels [27]. Because of the immobility of the patient during and after surgery, there may be a greater risk of blood clots as a result of the procedure. Advanced age, obesity, smoking, and previous history of blood clots are additional risk factors for cancer-related thromboembolism and stroke [3].

Treatment and assessment guidelines for cancer-related stroke or thromboembolism need to be developed to benefit this subgroup of stroke patients, given its unique aetiopathophysiology distinct from other subtypes of strokes [3]. Atherosclerosis, nonbacterial thrombotic endocarditis, disseminated intravascular coagulation, infection, tumour embolism, and thrombosis of the longitudinal venous sinuses are the most common mechanisms of cerebral ischaemia in the cancer patient [61]. Intratumoural bleeding, hypertensive haemorrhage, and coagulopathy are the most common causes of brain haemorrhage [61]. Notably, in cancer patients, an acute intracerebral haemorrhage can potentially be the initial manifestation of an underlying haematological disorder [62]. Haematologic intracerebral haemorrhages should be distinguished from other aetiologies of haemorrhagic strokes as they may have varying prognoses and require different treatment approaches for optimal secondary prevention for recurrent cerebral vascular disease [62]. Concerning recommendations for future direction, the improvement and standardisation of screening tools are of utmost priority in bridging the gap between low- and middle-income countries (LMICs) around the globe [2,35]. In the therapeutic management of stroke in cancer patients, we provide a stratified toolkit for clinical decision making or management for treating clinicians, to inform their clinical decision making in determining the appropriate preventive therapy. Various steps based on our current understanding of the optimal treatment of stroke in cancer patients are outlined in Figure 1.

Overall, recent recommendations show LMWH and DOACs to be the most viable options to decrease thrombotic risk in cancer patients [44]. However, this decision must be carefully considered by the treating physicians as individual patients will have varying levels of bleeding and clotting risks based on previous medical history, cancer phenotype, patient preferences, and existing drug regimens. Previously, LMWH has been accepted as the gold standard in treating cancer-associated VTE, but the advent of DOACs and their introduction to cancer cohorts has sparked increased interest in their role, and potential contraindications [63]. The hallmark 2003 CLOT (Randomised Comparison of Low-Molecular-Weight Heparin Versus Oral Anticoagulant Therapy for the Prevention of Recurrent Venous Thromboembolism in Patients with Cancer) trial demonstrated that patients on 6 months of LMWH (dalteparin) therapy had a decreased risk of recurrent VTE (9%) in comparison to those who received dalteparin treatment for 5–7 days followed by vitamin K antagonist (VKA) warfarin (17% risk) [63]. Another four randomised controlled trials comparing LMWH to warfarin have similarly found no statistically significant difference in the bleeding risk or overall mortality, thus rendering LMWH the predominant treatment of VTE in cancer patients for the last few decades [64,65,66,67].

Recommendations for DOAC use are mainly in low-risk ambulatory cancer patients who have an intact upper gastrointestinal tract and can comply with oral medication use and in hospitalised patients who do not have an impending surgery [44]. The NCCN recently released guidelines that state that DOACs, such as apixaban, edoxaban, and rivaroxaban, are preferred in patients without a history of luminal gastric or gastroesophageal lesions or recent peptic ulcer disease, with DOAC use associated with an increased risk of haemorrhage for patients diagnosed with gastrointestinal and genitourinary cancer [68]. Other cancer phenotypes, which are particularly vulnerable, include renal cell carcinoma, melanoma, thyroid cancer, and patients who have had recent (<1 month) brain surgery or who have metastatic brain lesions [44]. As such, we recommend that LMWH is preferred amongst this select cohort. DOAC usage should also be handled with caution in patients with renal impairment with a creatinine clearance < 30 mL/min and severe hepatic impairment with coagulopathy, at a Child–Turcotte–Pugh class B or C of cirrhosis [44]. Furthermore, DOAC use is contraindicated in patients on anticancer therapies that impact the P-glycoprotein, CYP3A4, or CYP2J2 pathways. This is corroborated by a systematic review and meta-analysis of 759 articles including 4 RCTs of 2894 cancer patients on either DOACs or LMWH [69]. It revealed that DOACs significantly reduced recurrent VTE incidence compared to LMWH, at 5.2% vs. 8.2%; however, they were associated with a nonsignificant higher bleeding risk at 4.3% vs. 3.3%, respectively, and a significantly higher increase in clinically relevant, nonmajor bleeding at 10.4% vs. 6.4%, respectively. Thus, although they are most efficacious in reducing VTE incidence, patients with a high bleeding risk should continue to be prescribed LMWH instead. The risk was again noted to be highest amongst the subgroup of patients with gastrointestinal or genitourinary cancer. It is of note, however, that rates of intracranial and fatal bleeds between DOACs and LMWH were comparable at 0.1% vs. 0.5% and 0.1% vs. 0.3%, respectively.

Finally, warfarin use, even given the significant bleeding risk, is indicated in patients who have a contraindication to DOACs and are intolerable to LMWH, and in patients with end-stage renal disease, nearing or on haemodialysis [44]. 

Regarding the duration of anticoagulant treatment, the NCCN guidelines recommend a minimum of 3 months; however, given that clinical trials of LMWH and DOACs assess efficacy over 6 months, it is perhaps safest to follow a 6-month minimum period. If the malignancy is still present, O’Connell et al. suggest continuing anticoagulant therapy beyond the 6 months until there is evidence of no disease, and the patient is in complete remission [44]. For patients with metastatic cancer or patients with a treatment regimen longer than 6 months, they recommend continuing anticoagulation beyond 6 months except when contraindicated, such as in cases with remission of haematologic malignancy.

The final consideration in prescribing DOACs vs LMWH is that of cost. In Latin American countries, LMWH holds a high pharmacy purchase cost, with the long-term regimen of subcutaneous LMWH injections posing a financial barrier for cancer patients of a lower socioeconomic background in obtaining the appropriate and optimal form of anticoagulation in preventing VTE [70,71]. In some cases, patients have been admitted to the hospital and started on unfractionated heparin via a continuous infusion pump to avoid the higher cost of outpatient bridging LMWH injections before VKA therapy [72] Alternatively, DOACs do not require constant laboratory monitoring of the anticoagulant effect and are thus a more accessible alternative both in terms of convenience and cost to LMWH injections. However, comparatively, warfarin is still the cheapest option, and thus in countries where cost is the main limiting factor, despite higher DOAC efficacy, it may remain the only realistic answer to VTE prophylaxis in cancer patients [72].

In summary, even though DOACs are an effective and accessible treatment option for cancer-associated VTE, the following various factors must be taken into account when using DOACs for the management of cancer-associated VTE [41,72,73,74]: (a) Increased risk of bleeding: Patients with renal impairment, for example, may be predisposed to an increased risk of bleeding from cancer-associated VTE due to underlying comorbidities and treatment modalities, warranting a careful selection of patients and monitoring for signs of haemorrhage [60]. (b) Limited clinical trial data: There is a paucity of clinical trial data on the efficacy and safety of DOACs in cancer-associated VTE, especially in those with certain types of cancer, such as gastrointestinal or genitourinary malignancies [44]. Therefore, it is essential to carefully examine each patient’s medical history and comorbidities. (c) Interactions with other medications/drugs: DOACS can interact with other medications that share the same metabolic pathway, as the DOACS are metabolised by the liver [75]. Drug interactions can be avoided with a careful review of the patient’s medication list. (d) Renal impairment: Individuals with cancer-associated VTE may experience renal impairment due to their underlying condition or treatment regimen [44,60]. Careful dosage adjustments and monitoring of renal function are necessary because the renal system accounts for a major portion of the elimination of DOACs. (e) Patient preference: Patients may have a preference for a certain treatment approach based on their own beliefs, lifestyle, and treatment goals [76]. Therefore, it is essential to have patients actively participate in decision-making processes and provide them with accurate information about the benefits and disadvantages of available treatments. (f) Cost and insurance coverage: DOACs may be costly, and insurance coverage may not be as comprehensive, depending on the patient’s insurance plan [72]. Consequently, when considering a treatment plan, it is important to take into account the patient’s financial status and insurance coverage. In conclusion, DOACs are an excellent treatment option for cancer-related VTE; however, careful consideration of patient-specific factors is required in order to ensure an effective and safe treatment course.

Prevention and management of cancer-related stroke or thromboembolism involve a multidisciplinary approach, encompassing oncologists, haematologists, and neurologists, and careful risk assessment, prophylactic measures, and prompt medical intervention when necessary [41,77]. The identification of patients who are at high risk for such events and the subsequent implementation of strategies to lower that risk are both examples of preventive measures. The treatment may consist of thrombolytic therapy, anticoagulant therapy (to prevent further clot formation), or mechanical procedures (such as stents or filters) to remove the clot or prevent the embolisation of the clot [43]. Because cancer patients are at a high risk for both ischaemic and haemorrhagic events due to the dysregulation of their haemostatic system by malignancy, determining the optimal antiplatelet drugs and their duration and dosage in those patients with a history of PCI, AIS, and/or ACS remains a challenge [51]. The use of cutting-edge technologies, such as drug-eluted stents (DESs) and OCT allows for the possibility of decreasing the length of DAPT in all patients, including those with cancer [53,78]. However, these technologies need further evaluation in large prospective trials or RCTs. Further research is warranted on developing pathways for stroke in cancer patients, particularly on the efficacy of reperfusion therapy for stroke in cancer patients, primary and secondary stroke prevention strategies in cancer, and fundamental research that can inform common mechanisms that predispose patients to an increased risk of stroke in cancer.

## Figures and Tables

**Figure 1 ijms-24-07981-f001:**
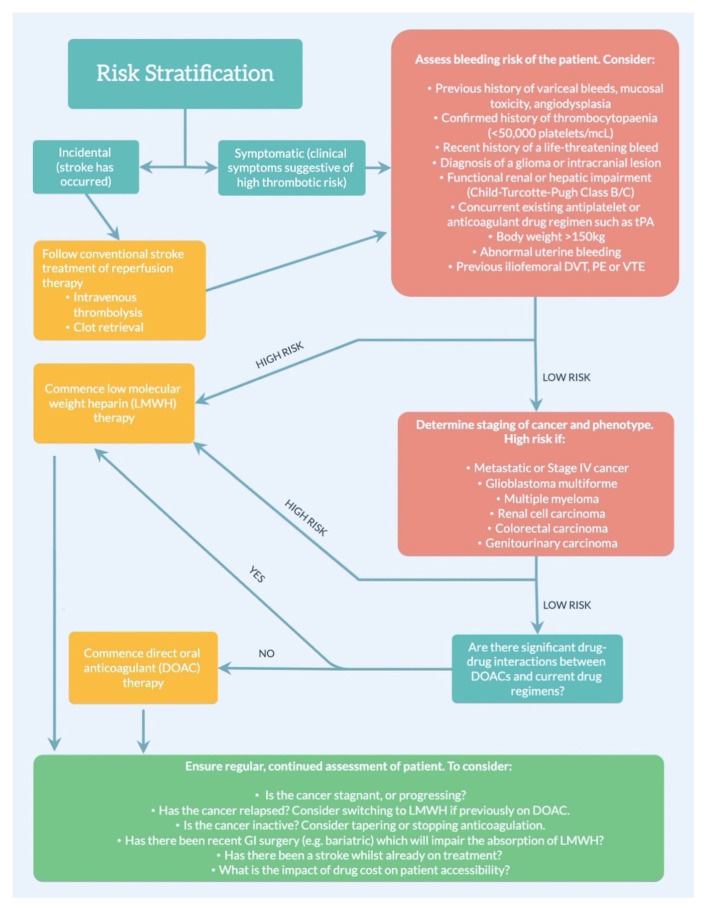
**Recommendations for future practice in the management of stroke in cancer patients.** Abbreviations: DVT: deep venous thrombosis; PE: pulmonary embolism; VTE: venous thromboembolism; DOAC: direct-acting oral anticoagulant; LMWH: low-molecular-weight heparin; GI: gastrointestinal.

## Data Availability

The original contributions presented in the study are included in the article, and further inquiries can be directed to the corresponding author.

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
