# Peer review of "Bridging the Gap in Cancer-Related Stroke Management: Update on Therapeutic and Preventive Approaches"

_ijms, 2023, doi:10.3390/ijms24097981_

Round 1

Reviewer 1 Report

Dear authors,

Your manuscript is an interesting effort to approach a very demanding, complex and wide topic. It is presented as a review (narrative I suspect) which is missing a lot of recent literature.

As such, I would recommend that you need to reconsider whether such an extensive topic can be covered in one manuscript and also review your search strategy and use of references to improve the quality of your work.

Some of the areas that need improvement are highlighted below.

11)      Introduction: Cancer related stroke is not exactly a ‘subtype’ of stroke. Please consider rephrasing.

22)     Introduction: The phrases ‘Though there are a variety of prophylactic treatments and guidelines available for the management of stroke in cancer’ and ‘the guidelines specific to the management and preventative screening of stroke in cancer patients are very limited’ sound contradictory. Please review.

33)     Therapeutic or Preventative Management of Stroke in Cancer Patients: ‘Overall, amongst both cancer and non-cancer patients, the proportion of incidental 57 ischaemic and non-ischaemic strokes remain consistent at 85% and 15% respectively’. Incidence is not the same as ‘incidental’ finding. I believe your reference refers to incidence rather than incidental identification of strokes. Please amend.

44)      Reperfusion therapy for stroke in cancer: Although you suggest this is a review article you have limited your comments and references on 3 studies, and have missed more recent evidence and large volume data. Examples:

-         ‘Immediate and Long-Term Outcomes of Reperfusion Therapy in Patients With Cancer’, Stroke. 2021;52:2026–2034.

-         ‘Mechanical Thrombectomy for Acute Ischemic Stroke in Patients with Malignancy: A Systematic Review’. J Clin Med. 2022 Aug; 11(16): 4696.

-         ‘Investigating Outcomes Post Endovascular Thrombectomy in Acute Stroke Patients With Cancer’. Neurology . 2022 Sep 19;10.1212/WNL.0000000000201208.

55)      Anticoagulation: Paragraph starting with ‘Some preliminary existing literature on this front from 2014 suggests that routine thromboprophylaxis is not ideal, only for high-risk patients [23,24].’ Again the authors fail to identify uptodate literature such as:

- ‘2022 international clinical practice guidelines for the treatment and prophylaxis of venous thromboembolism in patients with cancer, including patients with COVID-19’. Lancet Oncol. 2022 Jul;23(7):e334-e347.

- Venous Thromboembolism Prophylaxis and Treatment in Patients With Cancer: ASCO Clinical Practice Guideline Update. J Clin Oncol. 2020 Feb 10;38(5):496-520.

- American Society of Hematology 2021 guidelines for management of venous thromboembolism: prevention and treatment in patients with cancer. Blood Adv (2021) 5 (4): 927–974.

6) Due to the above issues the discussion cannot be reviewed as it is based on not uptodate evidence.

Author Response

We would like to thank the reviewer for the careful review of our work and feedback. We have made our best of efforts to incorporate the suggestions. A point-by-point rebuttal is provided below.

C#1: Dear authors,         

Your manuscript is an interesting effort to approach a very demanding, complex and wide topic. It is presented as a review (narrative I suspect) which is missing a lot of recent literature.

As such, I would recommend that you need to reconsider whether such an extensive topic can be covered in one manuscript and also review your search strategy and use of references to improve the quality of your work.

Some of the areas that need improvement are highlighted below.

Reply# Thank you for the remarks. We have made changes to the manuscript based on the feedback.

C#2: Introduction: Cancer related stroke is not exactly a ‘subtype’ of stroke. Please consider rephrasing.

Reply# Thank you for the remark. We have rephrased it at instances to clarify this, e.g., in the introduction.

"It has a unique aetiopathophysiology distinct from non-cancer-related strokes, warranting tailored/distinct treatment considerations [3]."

"Lack of treatment guidelines and assessment guidelines for cancer-related stroke or thromboembolism, therefore, needs to be developed to benefit this subgroup of stroke patients, given its unique aetiopathophysiology distinct from other subtypes of stroke [3]. 

C#3: Introduction: The phrases ‘Though there are a variety of prophylactic treatments and guidelines available for the management of stroke in cancer’ and ‘the guidelines specific to the management and preventative screening of stroke in cancer patients are very limited’ sound contradictory. Please review.

Reply# We have revised these sentences as below.

"Though there are a variety of prophylactic treatments and guidelines available for the management of stroke [8], research specific to the high-risk phenotype of cancer patients is limited, and therapies are only utilized and implemented to a limited degree due to a lack of awareness amongst clinicians regarding the risk of stroke in cancer [9].

Whilst guidelines exist for the secondary prevention of stroke and transient ischemic attack (TIA) [8], the guidelines specific to the management and preventative screening of stroke in cancer patients are limited [10]."  

C#4: Therapeutic or Preventative Management of Stroke in Cancer Patients: ‘Overall, amongst both cancer and non-cancer patients, the proportion of incidental 57 ischaemic and non-ischaemic strokes remain consistent at 85% and 15% respectively’. Incidence is not the same as ‘incidental’ finding. I believe your reference refers to incidence rather than incidental identification of strokes. Please amend.

Reply# Indeed, it was in reference to incidence. We have revised it accordingly, as below.

Overall, amongst both cancer and non-cancer patients, the incidence of ischaemic and non-ischaemic strokes remain consistent at 85% and 15% respectively [11]. 

C#5: Reperfusion therapy for stroke in cancer: Although you suggest this is a review article you have limited your comments and references on 3 studies, and have missed more recent evidence and large volume data. Examples:

-         ‘Immediate and Long-Term Outcomes of Reperfusion Therapy in Patients With Cancer’, Stroke. 2021;52:2026–2034.

-         ‘Mechanical Thrombectomy for Acute Ischemic Stroke in Patients with Malignancy: A Systematic Review’. J Clin Med. 2022 Aug; 11(16): 4696.

-         ‘Investigating Outcomes Post Endovascular Thrombectomy in Acute Stroke Patients With Cancer’. Neurology . 2022 Sep 19;10.1212/WNL.0000000000201208.

Reply# We thank the reviewer for the comments. We have now expanded the discussion for this topic as below, including the three references suggested.

Reperfusion therapy for stroke in cancer

Reperfusion therapy, intravenous thrombolysis (IVT), and endovascular throm-bectomy (EVT) are currently the mainstay of acute management of ischemic stroke pa-tients [14]. However, only limited studies exist on the efficacy of IVT or EVT in stroke patients with active cancer, with a relatively small sample size [15-19]. Hence, treatment guidelines for stroke in cancer patients are not well-established. The 2019 guidelines from the American Heart Association and American Stroke Association suggest that acute ischemic stroke patients with systemic cancer and reasonable (>6 months) life expectancy may benefit from IVT [20]. No such recommendations are available for EVT in patients with acute ischemic stroke and active cancer [21]. A sub-study of the MR CLEAN Registry provided Class I evidence that patients with active cancer undergoing EVT for AIS had significantly worse functional outcomes at 90 days relative to those without active cancer. Authors also reported an increased risk of recurrent stroke in cancer patients [17]. A study by Joshi et al found that the functional outcomes and mortality at 90 days of patients fol-lowing EVT for AIS with and without cancer were similar in a propensity-matched analysis [16]. Cancer patients, on the other hand, had a much greater risk of HT [16]. Analysis of SECRET (Selection Criteria in Endovascular Thrombectomy and Thrombolytic Therapy) registry on 1338 patients who underwent reperfusion therapy, comprising of 62 patients (4.6%) with active cancer, 78 patients (5.8%) with nonactive cancer, and 1198 patients (89.5%) with no history of cancer revealed similar adverse events and 24-hour neurological improvement in patients with active cancer relative to other groups [22]. However, patients with active cancer were linked to poorer long-term functional outcomes, vis a vis functional independence at 3 months and mortality at 6 months. A systematic review comprising of 18 retrospective studies on the safety and efficacy of MT in cancer patients revealed EVT is safe in cancer patients with acute ischemic stroke, however, with higher mortality at 90 days and lower 90-day functional independence [23]. Interestingly, a recent study leveraging the National Inpatient Sample (NIS), a large inpatient database from the United States, showed that, in contemporary medical practice settings, acute ischemic stroke patients with comorbid cancer or metastatic cancer treated with EVT have similar rates of intracranial hemorrhage and favorable/routine discharges as patients without cancer [24]. However, patients with metastatic cancer were associated with significantly higher rates of in-hospital mortality compared to patients without cancer [24]. Indeed, the efficacy of EVT in cancer patients presenting with stroke merits further investigation. Besides, efforts to reduce treatment delays in providing time-critical reperfusion therapies to acute stroke patients, including those with pre-existing cancer, are also needed [25,26].

C#6: Anticoagulation: Paragraph starting with ‘Some preliminary existing literature on this front from 2014 suggests that routine thromboprophylaxis is not ideal, only for high-risk patients [23,24].’ Again the authors fail to identify uptodate literature such as:

- ‘2022 international clinical practice guidelines for the treatment and prophylaxis of venous thromboembolism in patients with cancer, including patients with COVID-19’. Lancet Oncol. 2022 Jul;23(7):e334-e347.

- Venous Thromboembolism Prophylaxis and Treatment in Patients With Cancer: ASCO Clinical Practice Guideline Update. J Clin Oncol. 2020 Feb 10;38(5):496-520.

- American Society of Hematology 2021 guidelines for management of venous thromboembolism: prevention and treatment in patients with cancer. Blood Adv (2021) 5 (4): 927–974.

Reply# We thank the reviewer for the suggestion. We have updated the section as below.

“Anticoagulation

The preventative and therapeutic management of stroke in cancer patients is es-sential in improving prognosis and in reducing the burden that these diseases indi-vidually and combined place on the health system.

Cancer patients undergoing chemotherapy may be at increased risk of VTE [27,28]. Identifying patients at high risk of VTE are crucial for prophylactic treatment strategies such as anticoagulation. Two landmark randomized clinical trials (RCTs) indicated that the VTE prophylaxis with direct oral anticoagulants (DOACs), following risk assessment, significantly reduced rate of VTE during chemotherapy [29,30]. The 2021 guidelines from the American Society of Hematology recommend stratification of cancer patients according to their VTE risk prior to chemotherapy, as well as considering patient-specific factors, using the Khorana risk score, which takes into account the cancer phenotype [31]. More recently, 2022 international clinical practice guidelines published in Lancet Oncology by the International Initiative on Thrombosis and Cancer (ITAC) also provided indications for treatment and prophylaxis of cancer-associated thrombosis [32]. The following are key recommendations (grade 1A or 1B): (1) low-molecular-weight heparins (LMWHs) for in-itial (first 10 days) and maintenance therapy of cancer-associated thrombosis; (2) in the absence of significant drug-drug interactions or gastrointestinal absorption impairment, direct oral anticoagulants (DOACs) are indicated for the initial treatment and maintenance of cancer-associated thrombosis in patients who are not at elevated risk of gastrointestinal or genitourinary bleeding; (3) use of LMWHs or DOACs to treat cancer-related thrombosis for a minimum of 6 months; (4) prophylactic treatment with LMWHs for an extended period of time (4 weeks) to prevent postoperative VTE after major abdominopelvic sur-gery, if there is no significant bleeding risk; and (5) use of LMWHs or direct oral anti-coagulants (apixaban or rivaroxaban) is indicated for primary prevention of VTE in ambulatory patients with locally advanced or metastatic pancreatic cancer undergoing anticancer treatment and have a low likelihood of bleeding [32].

The American Society of Oncology (ASCO) Clinical Practice guidelines also provide in-dications for prophylaxis and treatment of VTE in patients with cancer [33]. It indicated pharmacologic thromboprophylaxis to hospitalized patients with active malignancy in the absence of bleeding or other contraindications. Whilst routine pharmacologic thrombo-prophylaxis was not indicated to all outpatients with cancer; it recommended thrombo-prophylaxis with apixaban, rivaroxaban, or LMWH in high-risk outpatients with cancer (Khorana score of 2 or higher prior to starting a new systemic chemotherapy regimen). Besides, it also indicated pharmacologic thromboprophylaxis with either unfractionated heparin (UFH) or LMWH to all patients with malignant disease undergoing surgical intervention.

Patients with cancer and VTE should receive anticoagulation for a minimum of 6 months [34]. However, the National Comprehensive Cancer Network (NCCN) guidelines rec-ommend a minimum duration of 3 months. In patients with cancer associated VTE, a meta-analysis of randomized trials indicates a decreased risk of recurrent VTE albeit with a higher risk of bleeding compared to LMWH [35].”

C#7: Due to the above issues the discussion cannot be reviewed as it is based on not uptodate evidence.

Reply# The revised manuscript addresses all the concerns raised. Furthermore, the recent guidelines have also been discussed.

Reviewer 2 Report

This review article by Sun et al. focused on the prevention and management of cancer-associated stroke or thromboembolism. Authors reviewed literatures and mainly demonstrated anticoagulation therapy for the prevention and treatment for cancer associated stroke or thromboembolism. As authors mentioned, stroke and/or thromboembolism are major problem in patients with cancer. Therefore, the concept of this review article is valuable and the contents seem agreeable. Although this manuscript seems written well, authors may want to consider several minor issues as follows.

Minor comments

1) All abbreviations such as AIS, MR CLEAN, HT, RCT, LMWH, DVT, NOAC, INR, VKA, AF, and LMICs should be spelt out at the first time of use.

2) It seems unclear whether or not the management is different depending on the kind of cancer.

Author Response

C#1: This review article by Sun et al. focused on the prevention and management of cancer-associated stroke or thromboembolism. Authors reviewed literatures and mainly demonstrated anticoagulation therapy for the prevention and treatment for cancer associated stroke or thromboembolism. As authors mentioned, stroke and/or thromboembolism are major problem in patients with cancer. Therefore, the concept of this review article is valuable and the contents seem agreeable. Although this manuscript seems written well, authors may want to consider several minor issues as follows.

Reply# We thank the reviewer for the review of our work.

C#2: 1) All abbreviations such as AIS, MR CLEAN, HT, RCT, LMWH, DVT, NOAC, INR, VKA, AF, and LMICs should be spelt out at the first time of use.

Reply# As suggested, we have now defined all the abbreviations the first time they are mentioned.

C#3: 
2) It seems unclear whether or not the management is different depending on the kind of cancer.

Reply# Indeed, the management of patients must factor in the phenotype of cancer. For example, GI and GU cancers are linked with an increased risk of major bleeding with DOACs, DOACs should be avoided in these patients or used with caution. 
